# Development of Melt-Castable Explosive: Targeted Synthesis of 3,5-Dinitro-4-Methylnitramino-1-Methylpyrazole and Functional Derivatization of Key Intermediates

**DOI:** 10.3390/molecules30132796

**Published:** 2025-06-28

**Authors:** Elena Reinhardt, Lukas Bauer, Antonia H. Stadler, Henrik R. Wilke, Arthur Delage, Jörg Stierstorfer, Thomas M. Klapötke

**Affiliations:** 1Department of Chemistry, Ludwig-Maximilians-University of Munich, Butenandtstr. 5–13, 81377 Munich, Germany; elrech@cup.uni-muenchen.de (E.R.); luauch@cup.uni-muenchen.de (L.B.);; 2Eurenco, 84700 Sorgues, France; a.delage@eurenco.com

**Keywords:** melt-castable explosives, pyrazole, compatibility, bomb calorimetry, SSRT, toxicity

## Abstract

The problems associated with TNT necessitate the development of novel melt-castable compounds with melting points between 70 and 120 °C, a crucial endeavor in the field of energetic materials. This study introduces a promising melt-castable explosive based on nitropyrazole, whose melt-castable properties were achieved by the introduction of methyl groups. The synthesis of 3,5-dinitro-4-methylnitramino-1-methylpyrazole involves a three-step process starting from 3,5-dinitro-4-chloropyrazole, including substitution, nitration, and methylation reactions. Additionally, two alternative synthesis routes and six energetic salts were examined. Structural elucidation was conducted using conventional methods such as NMR, IR, and X-ray, while the energetic properties of the compound, including thermal behavior, sensitivities, and theoretical performance, were investigated. Also, compatibility with common explosives was investigated, the experimental enthalpy of formation by bomb calorimetry was determined, and an SSRT test was performed. Furthermore, the melt-cast explosive underwent an Ames test in order to assess its toxicity.

## 1. Introduction

The class of industrially used melt-castable energetic materials consists of only a few compounds, most prominently trinitrotoluene (TNT), yet they are of fundamental importance in energetic material military and industrial applications including warheads, artillery shells, and civilian demolition charges [1,2]. TNT remains the industry standard, forming the basis of key formulations such as Composition B (RDX, TNT), Octol (HMX, TNT), and Pentolite (PETN, TNT) (Figure 1) [1,3]. However, its synthesis generates undesired asymmetrical TNT derivatives that must be removed [4]. This is typically achieved through washing with an aqueous sodium sulfite solution, known as the Sellite process, which produces large volumes of toxic, carcinogenic, and mutagenic wastewater, referred to as red water [5,6,7,8].

For large-scale TNT production, either an effective treatment method or a strategy to prevent red water formation is required [5]. However, treatment is costly and poses environmental risks, while process modifications to eliminate red water generation can compromise production efficiency [7]. A viable alternative is the development of new melt-castable energetic materials that avoid these issues altogether.

One of the primary criteria for a TNT replacement is thermal properties. A suitable substitute must exhibit a melting point between 80 and 120 °C, allowing the use of a melt-casting technique over a water steam bath. Moreover, its decomposition temperature should exceed the melting point by at least 100 °C, significantly reducing the risk of accidental ignition or decomposition during the melting process [9,10]. Another critical aspect is the energetic performance of the replacement material. The new compound should provide comparable or superior detonation velocity and pressure to maintain its effectiveness as an explosive. This ensures that the replacement not only matches but potentially exceeds the performance of TNT, making it viable for military and industrial applications. Safety is also a paramount consideration [9]. The alternative compound must be less sensitive to impact, friction, and shock, minimizing the likelihood of accidental detonation during handling, transportation, or storage [9]. Environmental and health concerns are also key drivers in the development of TNT alternatives. TNT is known for its persistent environmental contamination and toxicological impact on human health. A promising replacement should be non-toxic or exhibit significantly reduced toxicity and present a lower risk of contaminating soil and water ecosystems [11]. Compatibility with other explosives and additives is another fundamental requirement for TNT replacements. Since explosives are rarely used as pure substances in practical application, they are typically combined with other energetic materials (e.g., RDX, HMX) and functional additives such as binders, plasticizers, or metallic fuels. Compatibility testing is necessary to confirm that replacement material maintains its integrity and does not negatively impact the properties of the overall formulation [12]. Together, these criteria—thermal stability, energetic performance, safety, environmental sustainability, and compatibility—define the key parameters for the successful development of a new TNT replacement.

Nitropyrazoles are well suited as a scaffold for melt-castable explosives due to their favorable properties. Their high nitrogen content affords elevated heats of formation, whereas the four substitution sites on the pyrazole ring permit systematic property tuning [13,14]. A prominent example is 3,4-dinitropyrazole (DNP), which has been identified as a promising melt-castable matrix. DNP demonstrates a melting point (71 °C) [15,16] conducive to melting processes and offers improved detonation performance (8426 m s^−1^) [15] compared to TNT. However, due to its higher viscosity and acidic NH function, the careful adjustment of the formulation is required to ensure suitable workability [17,18]. Our previous work has described the nitroalkyl and azidoalkyl derivatives of different nitropyrazoles as potential melt-castable explosives [19]. In this study, we want to introduce 3,5-dinitro-4-methylnitramino-1-methylpyrazole, patented by EURENCO [20], as a prospective TNT replacement. In addition to a comprehensive thermal and physicochemical analysis, we investigated its experimental heat of formation, compatibility with common energetic co-formulants, quantified its performance in the Small-Scale Reactivity Test (SSRT), and assessed its toxicity.

## 2. Results and Discussion

### 2.1. Synthesis

In the current synthesis protocol, the intermediate 4-chloro-3,5-dinitropyrazole (**1**), synthesized via pyrazole chlorination [21] followed by subsequent nitration [22,23], is converted into 3,5-dinitro-4-methylaminopyrazole monohydrate (**2**). This conversion is facilitated using an aqueous solution of methylamine within an optimized synthetic route [23,24]. Additionally, the anhydrous form of this compound can be obtained by desiccating the synthesized product at 50 °C overnight. This desiccation process results in a chromatic transition from yellow to orange. The methylamino group was subjected to nitration to produce 3,5-dinitro-4-methylnitraminopyrazole (**3**), following the procedure established by Dalinger [23]. The nitration was carried out using trifluoroacetic acid as the solvent and acetyl nitrate as the nitration agent, obtaining a high yield of 96%. Starting from **3**, a series of six ionic compounds (**3a**–**3f**) was formed through an acid–base reaction. To preserve the salts in the solid state, it was necessary to omit the use of water as a solvent. When water was used as a solvent for salt formation, it resulted in either the formation of a sticky mass or an extended time for solidification. In the concluding step, methylation was carried out with a yield of 69% by deprotonating **3** using hydrogen carbonate in water, followed by a reaction with dimethyl sulfate. The advantage here is the precipitation of 3,5-dinitro-4-methylnitramino-1-methylpyrazole (**4**) from the aqueous solution, enabling straightforward filtration. In cases where **4** fails to precipitate, extraction with ethyl acetate can be employed as an alternative method (Figure 2).

As an alternative approach, an effort was made to introduce the methyl group at an earlier stage (Figure 2). Compound **1** underwent successful methylation with dimethyl sulfate, resulting in 67% yield of 4-chloro-3,5-dinitro-1-methylpyrazole (**5**), which was further reacted with the aqueous methylamine solution, yielding 4-aminomethyl-3,5-dinitro-1-methylpyrazole (**6**) with 75% yield. Additionally, **6** can be obtained in a satisfactory yield of 80% through the methylation of compound **2** [25]. The final step, involving the nitration of the methylamino group to produce the targeted compound **4**, presented challenges within this alternative synthesis. The selected nitration conditions prevented the obtainment of compound **4** in a solid or powder form; instead, it was consistently obtained as a sticky mass. Therefore, it is advisable to reserve the methylation for the final step, as the resulting product precipitates from the aqueous neutral medium.

### 2.2. Crystal Structures

Appropriate crystals for compounds **3**, **3a**–**3f** and **4**–**6** were obtained through recrystallization (**3**, **5**: ethanol; **3b**, **3d**, **3f**: water; **4**: acetonitrile; **6**: methanol) or directly from the reaction mixture (**3a**, **3c**, **3e**). The details of the X-ray measurements and refinements can be found in the Appendix A. Additional information on the X-ray structure determinations has been deposited in the CCDC database with the following reference numbers: 2334166 (**3**), 2334159 (**3a**), 2334160 (**3b**), 2334157 (**3c**), 2334161 (**3d**), 2334158 (**3e**), 2334165 (**3f**), 2334164 (**4**), 2334163 (**5**), and 2334162 (**6**).

The ellipsoids, representing non-hydrogen atoms in all structures, are illustrated at the 50% probability level.

Figure 3 displays the crystal structures of the six salts of 3,5-dinitro-4-methylnitraminopyrazole (**3**). None of these compounds include water molecules in their structures. In general, the pyrazole anions in all salt crystal structures (**3a**–**3f**) presented here have nearly identical geometries. The N−N and C−N bond lengths within the pyrazole are all sustainably shorter than C−N single bonds (1.47 Å) but significantly longer than C=N double bonds (1.22 Å). This common feature explicitly demonstrates the delocalization of the negative charge over the aromatic ring system. Another important point to mention is that π-staggered arrangements have only been observed in the potassium and ammonium salts, where hydrogen bonding is of minor importance. In contrast, structures that are rich in hydrogen bonding, such the guanidinium salt **3b** or hydroxylammonium salt **3f**, are strongly determined by these intermolecular interactions.

The molecular unit cells and the extended structures of the two neutral compounds, **3** and **4**, are illustrated in Figure 4. Both pyrazoles crystallize in the monoclinic space groups *P*21 (**3**) and *P*21/c (**4**) with two and four molecules per unit cell, respectively. In both structures, it is observed that the nitro groups at the C1 and C3 atoms are only slightly twisted out of the plane of the pyrazole ring. However, the nitro groups of compound **4** exhibit a greater degree of twisting (NO_2_−C1 17.28°, NO_2_−C3 19.13°) compared to those of compound **3** (NO_2_−C1 0.92°, NO_2_−C3 10.17°) all in the same direction. This results in a different rotation of the methylnitramino group, which needs to be less twisted out of the pyrazole plane in compound **4** (53.27°) than in compound **3** (69.31°) due to the stronger twisting of the nitro groups. Additionally, the methyl group on the pyrazole ring of compound **4** protrudes slightly from the pyrazole plane (174.04–180°). The main difference between the two structures is that no intermolecular hydrogen bonds are formed in compound **4** compared to compound **3**. Compound **3** exhibits hydrogen bonds with a length of 2.196 Å between the proton on nitrogen N1 and the nitrogen N2 of the neighboring pyrazole, leading to the formation of regularly stacked layers. This observation likely contributes to the higher density observed in compound **3** (1.774 g/cm^3^ @ 109 K) compared to compound **4** (1.696 g/cm^3^ @ 102 K).

### 2.3. NMR Spectroscopy

NMR spectra, including ^1^H, ^13^C, and ^14^N, were recorded for all compounds presented in this work. The comparison of the neutral compounds **2** and **3** reveals a downfield shift in the methyl group (**2**: ^1^H 3.03 ppm, ^13^C 33.4 ppm; **3**: ^1^H 3.36 ppm, ^13^C 40.5 ppm), attributed to the deshielding effects induced by the neighboring nitro group. Upon the further methylation of the pyrazole ring (compound **4**), the methyl group attached to the pyrazole undergoes stronger deshielding (^1^H 4.35 ppm, ^13^C 43.2 ppm) compared to the methyl group attached to the nitramine (^1^H 3.35 ppm, ^13^C 40.2 ppm). The methyl group on the nitramine of compound **4** is slightly more shielded compared to **3**, resulting in its signal overlap with signals from DMSO-*d6* when recording the ^13^C spectrum in DMSO-*d6* at low concentrations. To clearly distinguish all methyl groups in the ^13^C spectrum, even at lower concentrations, the NMR spectrum can also be recorded in acetone-*d6*. The greater shift of the methyl group on the pyrazole ring compared to that on the nitramine is further confirmed by compounds **5** (^1^H 4.28 ppm, ^13^C 43.5 ppm) and **6** (^1^H 4.16 ppm, ^13^C 43.0 ppm).

Additionally, for compound **4**, ^15^N and ^1^H ^15^N HMBC NMR spectra were obtained. The ^15^N ^1^H-decoupled spectra of compound 4 reveal only five signals instead of the expected six. To locate the missing signal, the ^1^H ^15^N-coupled spectrum is employed (Figure 5). Focusing on the signal at −33 ppm reveals the presence of two nitro groups. The nitro group N6 induces a quartet splitting pattern by coupling with the methyl group. A singlet appears precisely at the center of the quartet, indicative of the nitro group N4. Additionally, the third nitro group N5 appears as a singlet at −27 ppm. The remaining three singlets each exhibit quartet splitting. The observed splitting in the pyrazole signals is attributed to the coupling with the methyl group on the pyrazole ring, while that of the N3 signal stems from the coupling with the methyl group on the nitramine. Signal assignment was accomplished utilizing the ^1^H ^15^N HMBC spectrum.

### 2.4. Physiochemical Properties

The physicochemical properties, including thermal behavior, sensitivities to impact, friction, and electrostatic discharge, as well as the detonation parameters of the six salts of compound **3** and neutral compounds **3**, **4,** and **6**, are summarized in Table 1 (more detailed tables can be found in the Appendix A).

A comparison of the salts with each other and with the neutral compound **3** reveals different trends. The potassium salt (**3a**) has the highest thermal stability (217 °C) and density at room temperature (1.906 g/cm^3^), but has the lowest detonation velocity (7296 m/s). Conversely, the hydroxylammonium salt (**3f**) has the highest detonation velocity of 8470 m/s and a commendable density of 1.724 g/cm^3^, albeit with the lowest decomposition point at 147 °C. Only the guanidinium (**3b**) and aminoguanidinium (**3c**) salts show an endothermic event just before decomposition. In terms of decomposition temperatures, all salts, with the exception of potassium (217 °C), show a similar range (147–169 °C) compared to the neutral compound **3** (157 °C). However, the formation of salts (excluding potassium and hydroxylammonium salts) leads to a decrease in density compared to the neutral compound. Regarding sensitivities, the guanidinium salt (**3b**) is characterized by an insensitivity towards impact and friction (confirmed twice with different batches). Conversely, the other salts show an increased sensitivity to impact compared to the neutral compound **3**. While only the potassium (**3a**) and hydrazinium (**3e**) salts show a higher sensitivity to friction compared to the neutral compound, the others show a lower sensitivity.

Among the neutral compounds, compound **3** is characterized by the highest density (1.725 g/cm^3^) and detonation velocity (8228 m/s). However, it has no melting point and exhibits a relatively low decomposition temperature of 157 °C. It also shows increased sensitivity to impact (8 J) and friction (144 N). The methylation of compound **2** on the pyrazole ring yields compound **6**, a melt-castable derivative with an acceptable decomposition temperature of 203 °C and insensitivity to mechanical stimuli. Nevertheless, its melting point is somewhat too high for practical applications (133 °C) and its detonation velocity of 7273 m/s is rather low. The combination of compounds **2** and **3** to form compound **4** leads to a promising TNT replacement. Compound **4** has a sensitivity (IS: 15 J, FS: >360 N) and a density (1.648 g/cm^3^) comparable to TNT and a suitable melting point of 77 °C (Figure 6). Although its decomposition temperature (190 °C) is lower than that of TNT (289 °C) or DNAN (315 °C), the necessary safety margin of at least 100 °C between the melting and decomposition point is maintained. It is significant that compound **4** has a detonation velocity of 7721 m/s and a detonation pressure of 24.1 GPa, both of which are higher than those of TNT or DNAN. Since the heat of formation influences the detonation parameters [1], it was additionally determined experimentally for compound **4** using a bomb calorimetry (details on the measurement can be found in SI, S4 Bomb Calorimeter). A comparison between the experimental (106.4 kJ/mol) and calculated values (135.2 kJ/mol) reveals that the calculation yielded a slightly more endothermic value (Table 1). The recalculation of the detonation parameters using the experimental heat of formation therefore resulted in slightly lower detonation velocity (7682 m/s) and pressure (23.8 GPa).

Compound **4**, recognized as a potential alternative to TNT, underwent compatibility testing with RDX, HMX, CL-20, and TKX-50 (Figure 6). Compatibility measurements were performed using differential thermal analysis (DTA). The results demonstrate a high degree of compatibility between compound **4** and both RDX and CL-20, characterized by a minimal difference of 0 to 4 °C between their melting points and decomposition temperatures. HMX and TKX-50 demonstrate a deviation of 1 °C only at the melting point. However, compatibility for HMX at the decomposition temperature is rated moderate (6 °C difference), albeit close to the lower limit of good compatibility. TKX-50 demonstrates incompatibility at the decomposition point with compound **4**, as indicated by a temperature difference of 23 °C (for additional information, see SI, S8 Compatibilities).

### 2.5. Small-Scale Shock Reactivity Test

The SSRT test evaluates the explosiveness of compounds on a small scale of ~500 mg and a fixed volume of 283 mm^3^. An explosive compound is filled into a steel cylinder with a borehole and pressed on top of an aluminum witness block. The compound is filled into the hole and pressed with a weight of 3 t. A commercial detonator is used to initiate the explosion, creating a dent in the aluminum block. The dent volume can be measured using optical topography and can then be compared to other energetic materials [27]. Several factors such as overall energy release, detonation velocity, and pressure are combined in this test setup to assess the compound’s general explosiveness. The tested compound **4** significantly outperforms TNT by 22% in dent volume, although the density of compound **4** is the same (Table 2). The values are still comparable to PETN, which has a higher density than compound **4** (for addition information see SI, S7 SSRT).

### 2.6. Ames Test

Performing an Ames test on a new explosive is essential to identify potential risks arising from exposure to that substance. Given the wide range of applications for explosives in the military and civilian sectors, it is imperative to ensure if genetic mutations and associated health risks to both humans and the environment occur. Carrying out an Ames test enables the early detection of potential hazards and facilitates the implementation of appropriate measures to ensure human and environmental safety. The test uses specific bacterial strains with genetic mutations that prevent them from producing certain amino acids. When a substance with mutagenic potential is tested, it can reverse these mutations, leading to bacterial growth. The mutagenic activity of the tested substance can be concluded by observing the extent of bacterial growth (for more information, see SI, S9 Ames Test).

Compound **4** showed a mutagenic effect in all strains in both the assay without and with metabolic activation. The increased number of revertant colonies in the assay without metabolic activation was observed at the concentrations of 4.11–0.46 μg/plate for TA98 and TA1537 strains, 1.37–0.46 μg/plate for TA100 and TA1535, and 37.03–0.46 μg/plate for *E. coli*. After metabolic activation, the number of revertant colonies increased for TA98 at concentrations ranging from 12.3 to 0.4 μg/plate, for TA100 from 4.11 to 0.46 μg/plate, for **TA1535** from 4.11 to 1.37 μg/plate, for TA1537 from 12.3 to 1.37 μg/plate, and for *E. coli* from 37.03 to 4.11 μg/plate.

## 3. Materials and Methods

All used materials and methods and the detailed experimental part can be found in Appendix A.

## 4. Conclusions

In this study, six salts of 3,5-dinitro-4-methylnitraminopyrazole (**3**) were synthesized and characterized. The potassium salt showed a remarkable thermal stability, with a decomposition temperature of 217 °C. The hydroxylammonium salt (**3f**) displayed the highest detonation velocity of 8470 m/s. The guanidinium salt (**3b**) showed the desirable complete insensitivity to mechanical stimuli. The aminoguanidinium (**3c**), ammonium (**3d**) and hydrazinium (**3e**) salts showed properties very similar to those of the neutral compound (**3**). The most interesting new compound is the melt-castable energetic material 3,5-dinitro-4-methylnitramino-1-methylpyrazole (**4**). The substitution of acidic protons with methyl groups weakens intermolecular interactions just enough to result in a lowered melting point of 77 °C while still retaining sufficient thermal decomposition temperature of 190 °C. The introduction of the methyl group to the pyrazole ring in the final synthesis step yields the most optimal results, as the product readily precipitates from water. Precautions should be taken due to the observation of potentially mutagenic effects in the Ames test. Compound **4** demonstrates compatibility with RDX, HMX, and CL-20, indicating its potential application as a drop-in TNT replacement. The sensitivity to mechanical stimuli and density at room temperature of compound **4** are comparable to those of TNT, while surpassing TNT’s detonation velocity by more than 10%.

## Figures and Tables

**Figure 1 molecules-30-02796-f001:**
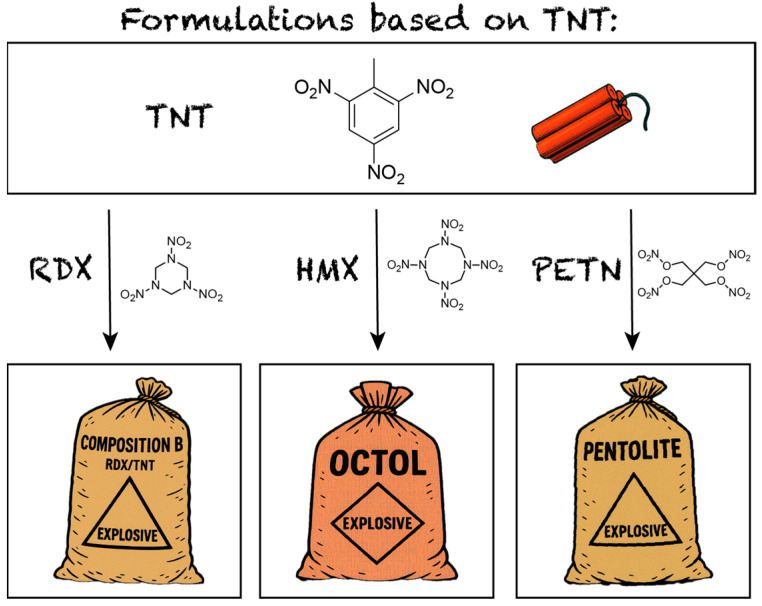
The overview of the most common formulations using TNT as the melt-castable component.

**Figure 2 molecules-30-02796-f002:**
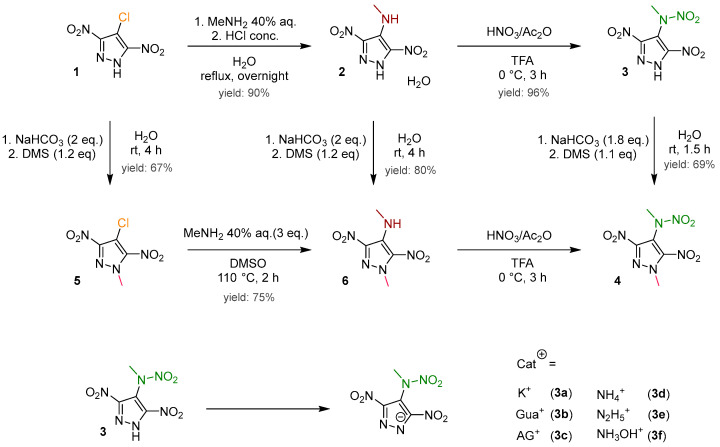
The synthesis overview of 3,5-dinitro-4-methylnitramino-1-methylpyrazole (**4**) and the salts of 3,5-dinitro-4-methylnitraminopyrzole (**3**).

**Figure 3 molecules-30-02796-f003:**
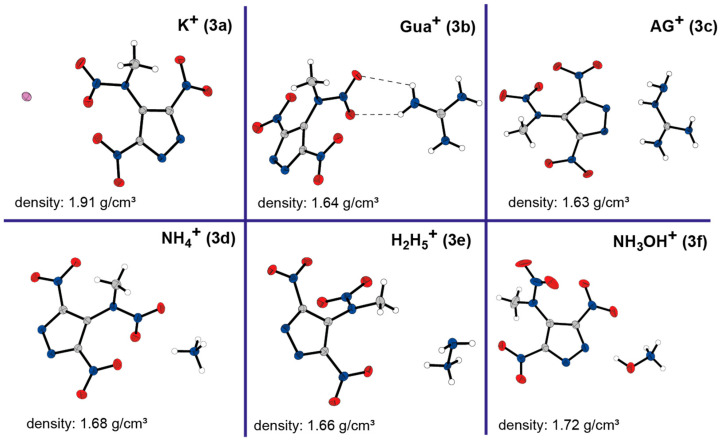
Molecular units of synthesized salts of compound **3**: **3a** potassium; **3b** guanidinium; **3c** aminoguanidinium; **3d** ammonium; **3e** hydrazinium; and **3f** hydroxylammonium.

**Figure 4 molecules-30-02796-f004:**
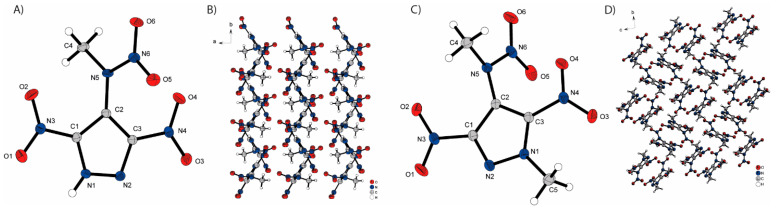
Molecular units of synthesized neutral compounds and its extended structure: (**A**) 3,5-dinitro-4-methylanitraminopyrazole (**3**), (**B**) extended structure of **3**, (**C**) 3,5-dinitro-methylnitramino-1-methylpyrazole (**4**) and (**D**) extended structure of **4**.

**Figure 5 molecules-30-02796-f005:**
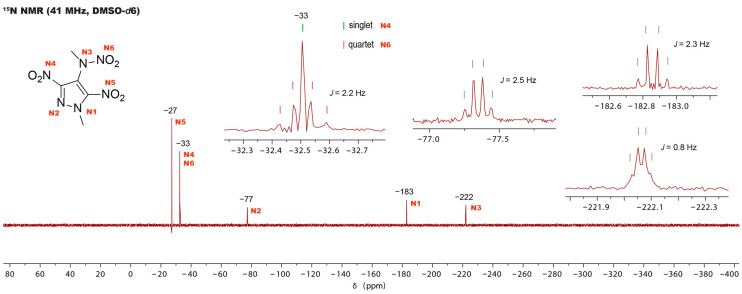
The proton-coupled ^15^N NMR spectrum of compound **4**, showing the splitting of the signals.

**Figure 6 molecules-30-02796-f006:**
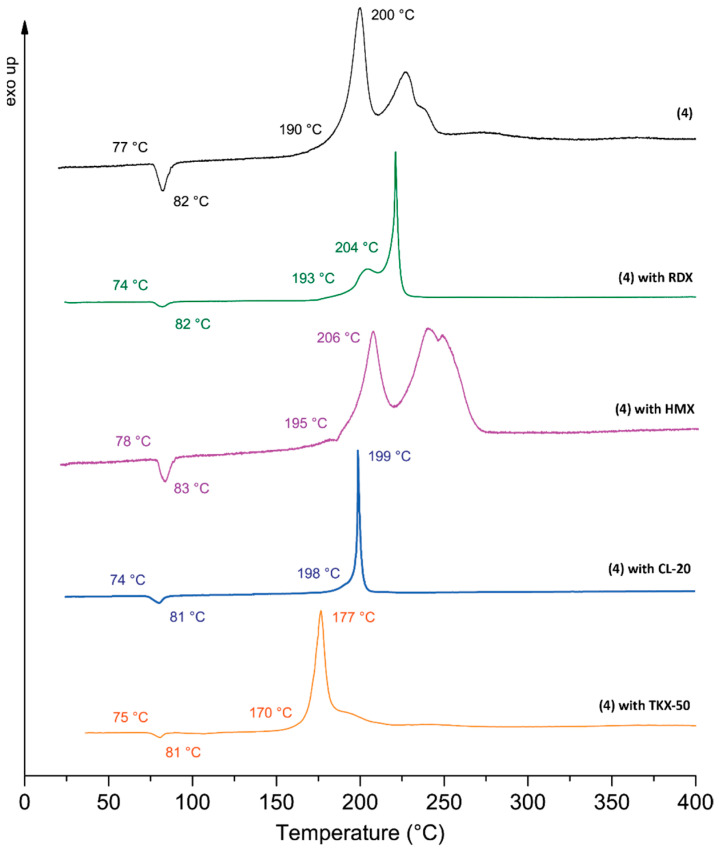
DTA compatibility measurements of **4** with RDX, HMX, CL-20, and TKX-50.

**Table 1 molecules-30-02796-t001:** Physiochemical properties and detonation parameters of all synthesized energetic compounds compared to TNT and DNAN.

	*IS* ^[a]^ [J]	*FS* ^[b]^ [N]	*T_endo_* ^[c]^ [°C]	*T_exo_* ^[d]^ [°C]	*Ρ* ^[e]^ [g cm^−3^]	Δ_f_*H°* ^[f]^ [kJ∙mol^−1^]	*D*_C-J_ ^[g]^ [m∙s^−1^]	*p*_C-J_ ^[h]^ [GPa]
**3a**	6	80	/	217	1.906	−446.5	7296	21.7
**3b**	>40	>360	140	158	1.638	60.0	7719	22.7
**3c**	8	168	116	148	1.628	165.7	7848	23.3
**3d**	3	192	/	169	1.681	101.9	8129	26.8
**3e**	5	120	/	165	1.660	248.5	8261	27.3
**3f**	5	192	/	147	1.724	157.7	8470	30.5
**3**	8	144	/	157	1.725	145	8228	28.8
**6**	>40	>360	133	203	1.629	76.1	7273	19.6
**4**	15	>360	77	190	1.648	135.2/106.4 ^[j]^	7721/7682 ^[j]^	24.1/23.8 ^[j]^
**TNT** ^[i]^	**15**	**>360**	**81**	**289**	**1.65**	**−185**	**6950**	**20.5**
**DNAN** [26]	**>40**	**>360**	**94**	**315**	**1.59**	**−177**	**6705**	**16.1**

^[a]^ Impact sensitivity (BAM drop hammer, method 1 of 6); ^[b]^ friction sensitivity (BAM friction tester, method 1 of 6); ^[c]^ endothermic event (DTA, β = 5 °C∙min^−1^); ^[d]^ temperature of decomposition (DTA, β = 5 °C∙min^−1^); ^[e]^ density at 298 K recalculated from X-ray data; ^[f]^ heat of formation (calculated using the atomization method and CBS-4M enthalpies); ^[g]^ detonation velocity; ^[h]^ detonation pressure; ^[i]^ determined at LMU; ^[j]^ determined experimentally (bomb calorimetry).

**Table 2 molecules-30-02796-t002:** Results of SSRT for **4** compared to PETN and TNT.

	4	PETN	TNT
Dent−Volume [mm^3^]	1037.34	1107.81	845.89
m [g] ^[a]^	443	478	443
*ρ* [g cm^−3^] ^[b]^	1.648	1.778	1.648
*p*_C-J_ [GPa] ^[c]^	24.1 ^[e]^	30.8 ^[e]^	19.4 ^[e]^
*V_det_* [m s^−1^] ^[d]^	7721 ^[e]^	8429 ^[e]^	6839 ^[e]^

^[a]^ Used mass; ^[b]^ density; ^[c]^ detonation pressure; ^[d]^ detonation velocity; ^[e]^ calculated with EXPLO5_V6.05.

## Data Availability

The data presented in this study are available in the article and Appendix A.

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
