# Peer review of "Development of Melt-Castable Explosive: Targeted Synthesis of 3,5-Dinitro-4-Methylnitramino-1-Methylpyrazole and Functional Derivatization of Key Intermediates"

_molecules, 2025, doi:10.3390/molecules30132796_

Round 1
Reviewer 1 Report
Comments and Suggestions for Authors
Comments on the manuscript
This study introduces a promising melt-castable explosive based on nitropyrazole, whose melt-castable properties were achieved by introduction of methyl groups. The synthesis of the title 3,5-dinitro-4-methylnitramino-1-methylpyrazole involves a three-step process starting from 3,5-dinitro-4-chloropyrazole, including substitution, nitration, and methylation reactions. A series of methods were used to study the energy characteristics, compatibility with common explosives and toxicity of the synthesized compounds and their ionic salts. Some new results were obtained, and the following modifications were suggested:
- In lines 75-76 of this article, verify the melting point and detonation velocity of DNP.
- Line 247 of the article does not match the information provided by the picture.
- The DTA method requires at least four DTA curves at different heating rates. Only one DTA curve is given in this paper, and it also did not specify the rate of heating
- In this paper, six kinds of 3,5-dinitro-4-methylnitropyrazole salts and 3,5-dinitro-4-methylnitro-1-methylpyrazole were synthesized and characterized. The research content of the paper does not match the title, it is suggested to modify the title.
- There are many substances that can replace TNT in the current research, why not compare with these substances?
Reviewer 2 Report
Comments and Suggestions for Authors
This MS described a promised melt-castable compound 3,5-Dinitro-methylnitramino-1-methylpyrazole (4), its properties were compared with TNT. The study provides a strategy of adding a methyl group. The melt point is only 77℃, which is lower than TNT. I just wonder, in the molecule of TNT, there is only one methyl group, and there are two methyl groups in the title compound, so what is the function of the methyl group? And really the more group of compound 4 than 3 makes the compound 3 has lower group, Anymore, the thermal decomposition peak temperature of 4 is much lower than TNT. Obviously, this is not good for the thermal stability, how to elucidate this?
The title compound is about 3,5-Dinitro-methylnitramino-1-methylpyrazole (4), however the derivates are from (3), so I donnot think the title is very suitable.
The thermal properties of TNT composite of RDX, HMX , etc should also provide for the comparison with the title compound.
2, 3, 4, they have similar skelton, I recommend the properties of 2 list in Table 1 for comparison and make deep discuss of this melt-castable compound.
The conclusion could be more concise.
In the Supplement doc. The DSC, TG and DTG we say curves never use spectrum.
Round 2
Reviewer 2 Report
Comments and Suggestions for Authors
All the issues are corrected, it can be accepted now.